# Direct Underwater Sound Velocity Measurement Based on the Acousto-Optic Self-Interference Effect between the Chirp Signal and the Optical Frequency Comb

Zihui Yang [1], Fanpeng Dong [2], Hongguang Liu [3], Xiaoxia Yang [4], Zhiwei Li [5] and Bin Xue [1,*]

1   School of Marine Science and Technology, Tianjin University, Tianjin 300110, China
2   School of Electrical and Information Engineering, Tianjin University, Tianjin 300110, China
3   Tianjin Institute of Metrological Supervision and Testing, Tianjin 300192, China
4   Tianjin Key Laboratory of Information Sensing & Intelligent Control, Tianjin University of Technology and Education, Tianjin 300222, China
5   Tianjin Key Laboratory for Control Theory and Applications in Complicated Systems, Tianjin University of Technology, Tianjin 300384, China
*   Correspondence: xuebin@tju.edu.cn

**Abstract:** Underwater sound speed plays a vital role in maritime safety. Based on the acousto-optic self-interference effect, we proposed a new method to measure underwater sound speed utilizing Raman–Nath diffraction, generated by the acousto-optic effect between an optical frequency comb and pulsed chirp signal. When the pulsed chirp travels between the measurement and reference arm in the experimental setup that we constructed, the same signal resulting from acousto-optic self-interference is produced. The time gap between the two identical signals represents the time interval. Thus, we can determine the time-of-flight using cross-correlation. The optical path difference between the two arms is double the flight distance of ultrasonic waves and can easily be obtained using femtosecond laser interferometry. The time gap and the distance can be used to measure sound speed. The experimental results show that the chirp signal improves the signal-to-noise ratio and expands the applicable time-of-flight algorithm. The waveform pulse width after cross-correlation is 1.5 μs, compared with 40 μs before. The time-of-flight uncertainty can achieve 1.03 ns compared to 8.6 ns before. Uncertainty of sound velocity can achieve 0.026 m/s.

**Keywords:** underwater sound velocity; chirp signal; time-of-flight; acousto-optic self-interference; maritime safety

## 1. Introduction

Sound is fundamental to many forms of marine research. Compared with other radiation waves, acoustic waves in seawater have the best performance, the longest transmission distance, and the highest efficiency [1]. Due to the conductivity of water, radio frequency signals, which are capable of traveling great distances in the air, are limited to short distances underwater [2]. Also, because of the scattering in the water, light can only penetrate a few hundred feet underwater. Thus, sound might be the best tool to explore the ocean. For instance, in underwater communication, the use of acoustic modems has met the requirements of high bandwidth and long-distance travel, making acoustic communication the most efficient method of data transport in the sea [3], and in underwater object detection, acoustic imaging uses the sound propagation in solids and elastic properties to image the objects [4]. These applications rely on accurate sound velocity measurement [5]. Therefore, sound velocity is a crucial metric in marine engineering.

Generally, the sound speed in seawater can be measured directly and indirectly. The indirect method, represented by the empirical formula method, refers to the measurement of the seawater temperature, salinity, and pressure in real-time and establish a fluid equation

to calculate the underwater sound speed. There are a range of formulas, such as the Chen–Millero formula [6], Del Grosso formula [7], and Wilson formula [8,9]. However, temperature, pressure, and salinity measurements have their own uncertainties in these formulas, and there is also uncertainty in the formulas themselves. The question of which formula is the most accurate is a controversial subject [10,11]. Meanwhile, the indirect method is far from the definition of sound velocity, violating measurement traceability demand. The direct method, represented by the sing-around method, calculates sound speed by measuring the flight time of pulsed ultrasonic waves within a known distance using piezoelectric transducers [12,13]. However, determining the known distance travelled by sound is not an easy task with this method, and this combination of "piezoelectric transducers and sing-around" has fatal weaknesses in error source elimination. First, the width of a sound pulse transmitted by transducers cannot achieve an ideal value; hence an infinitely narrow pulse in the time domain is unattainable [14]. This deficiency makes pinpointing the peak of a pulse difficult, particularly when there is distortion [15,16]. Secondly, the piezoelectric effect can barely show a clear starting point of vibration when the sound pulse arrives or goes off, which inevitably introduces a time error. In the current situation, the definition based the direct method requires the indirect method to calibrate.

To solve the dilemma of traceability and precision, our previous study proposed more direct methods for measuring sound speed based on the acousto-optic effect. First, we measure sound speed based on the pulsed acousto-optic effect based on the frequency comb and the square pulse [17]. In this method, the driving signal of the transducer is the square pulse, and the acousto-optic interaction signal is a 40-microsecond-wide pulse. The algorithm for processing the acousto-optic interaction signal is the threshold method. This method seems to have good traceability and precision, however, in maritime application scenarios, the acousto-optic interaction signal is easily disrupted by noise, resulting in a rise in measurement error [18]. Although the time-of-flight algorithm based on the pulse model has been described before, the algorithm is too complicated, and the existing acousto-optic signal action is not yet sufficiently clear, so an estimation model cannot be established [19,20]. These flaws make the method hard to apply in marine engineering [21]. Second, we measure sound speed based on the He-Ne laser acousto-optic effect and sing-around method [22]. This method has the same flaw mentioned above, the acousto-optic mark signal used to measure the time-of-flight is hard to obtain in the presence of noise. Both the "square pulse + threshold" combination and the "mark-signal + sing-around" combination have poor noise immunity. Consequently, applying the current acousto-optic methods in marine engineering remains challenging.

To fill the gap between the current methods and the requirements of marine engineering, we discovered the acousto-optic self-interference effect that occurs when an ultrasonic wave interacts with an optical frequency comb. Diffraction occurs when light propagates through an ultrasonic field. Theoretically, the distinction between different orders of diffracted light is obvious. However, instead of being ideal, the lasers we used during the experiment had a divergence angle, causing different orders of diffracted light to interfere with each other. We named it the "self-interference", and this "self-interference" signal contains information about the ultrasonic field. Specifically, the sound pressure, frequency of the sound wave, and the phase of the sound wave. In this study, we use the "self-interference" signal generated by a chirp signal to measure the sound velocity. This "self-interference" signal has the characteristic of broad bandwidth and high signal-to-noise ratio. We then sought to take the unique feature of the signal to improve the resolution and precision of time-of-flight data [23].

According to the Cramer–Rao lower error bound (CRLB), time-of-flight is related to the center frequency, signal-to-noise ratio, and effective bandwidth of the signal [24–26]. Enhancing effective bandwidth and signal-to-noise ratio not only improves the accuracy of time-of-flight measurement, but also achieves better noise immunity. Thus, in this study, we use the chirp signal (300 kHz–1 MHz) as the driving signal to obtain the chirp acousto-optic interaction signal. The chirp signal introduces better effective bandwidth [27]. We use the

cross-correlation algorithm instead of the threshold method to achieve a better denoising performance. By utilizing the combination of "chirp + cross-correlation", sound velocity with good traceability can be obtained, which makes it possible to apply acousto-optic measurement in real-time marine engineering.

This article is organized as follows: Section 2 introduces the measurement principle and experimental device in detail. Section 3 first establishes the experiment and sets out the results, then shows the comparison with SVP. Section 4 introduces the uncertainty analysis. Finally, Section 5 summarizes the main conclusion of this work with a brief overview of future improvements.

## 2. Measuring Principle and Experimental Setup

Figure 1 shows the principle of the experiment. Light is emitted from a femtosecond optical frequency comb and enters a Mach–Zehnder interferometer. Both the measurement arm and the reference arm pass through the water tank, and the path difference between the measurement arm and the reference arm is twice the distance travelled by sound $S$, which is also the distance between the two arms of the interferometer. As the sound passes through the light, a beat frequency signal with the same frequency as the sound will be produced, allowing flight time of the soundwave $P$ to be determined by cross-correlation. The sound speed can be calculated as:

$$v = \frac{S}{P} \tag{1}$$

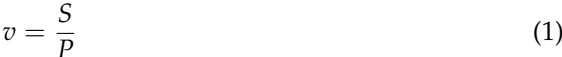

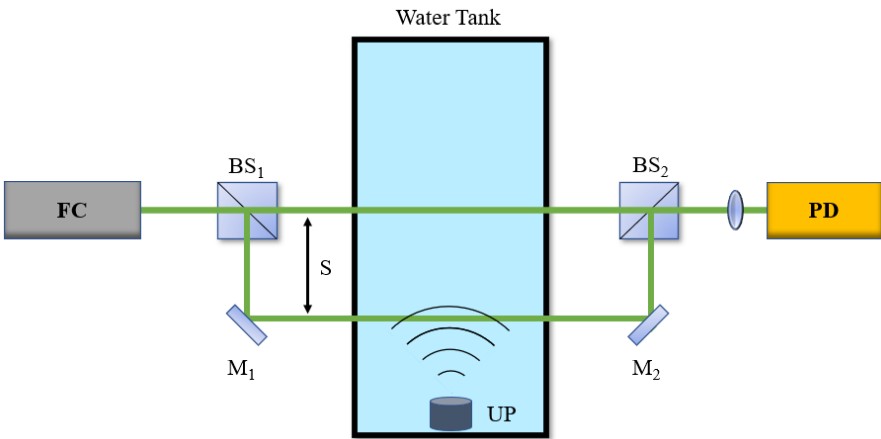

**Figure 1.** Experimental schematic diagram. FC: frequency comb; $M_{1-2}$: mirror; $BS_1$–$BS_2$: beam splitter; UP: ultrasound probe; PD: photodetector.

### 2.1. Principle for Time-of-Flight Measurement

The principle for time-of-flight measurement is based on the acousto-optic effect of a femtosecond optical frequency comb interacting with ultrasonic waves under Raman–Nath conditions [28]. Diffraction occurs when light travels through an ultrasonically perturbed material, accompanied by a doppler frequency shift [29]. The frequency of diffracted light at different orders changes at the original optical frequency by an integer multiple of the acoustic frequency (the multiple is numerically equal to the order of diffracted light). Figure 2 shows the acousto-optic action in the ideal case, the distinction between different orders of diffracted light is obvious.

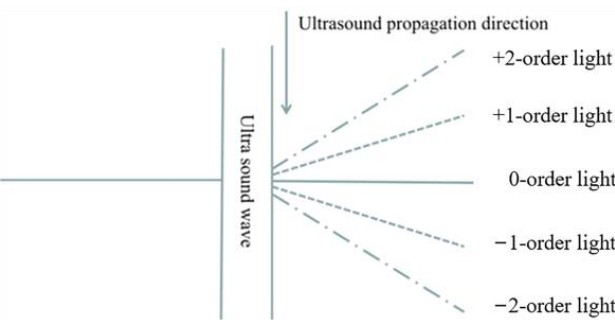

**Figure 2.** Principal diagram of acousto-optic action.

However, instead of being ideal, lasers have a certain divergence angle under experimental conditions, which cannot be ignored; take the He-Ne laser from Thorlabs (Thorlabs HRS015B) as an example. We take the lower limit of the divergency angle of the lasers, 1.4 mrad as the divergence angle. The calculation formula of the diffraction angle is as follow [30].

$$\sin(\theta_1) = \frac{n\lambda}{\lambda_\mu} \tag{2}$$

where $\lambda$ is the wavelength of the incident light, $\lambda_\mu$ is the ultrasonic wavelength in the medium and $\theta_1$ is the diffraction angle, $n$ is the order of the diffracted light. The frequency of the ultrasonic wave can be calculated as:

$$f_a = \frac{v_a \times \sin\theta}{\lambda} \tag{3}$$

where $v_a$ is the sound velocity, $\theta$ is the divergency angle and $\lambda$ is the wavelength of the laser. When $v_a$ is 1400 m/s, $\theta$ is 1.4 mrad, $\lambda$ is 632 nm, then the minimum frequency of distinct diffracted light is 3.3 MHz. In marine engineering, the frequency of the ultrasonic wave we use is considerably lower than the minimum frequency, then the diffracted light cannot be separated. Take the 0-order diffraction light and the 1-order diffraction light as an example. As shown in Figure 3, the solid line represents the 0-order diffraction light, whereas the dashed line represents the 1-order diffraction light. The diffraction angle is $\theta_1$ and the divergence angle is $\theta$; $\theta_1$ is considerably smaller than $\theta$. The formation of the upper edge light of 0-order diffraction light and the lower edge light of 1-order diffraction light remains divergent, so diffracted light cannot theoretically be separated. Thus, different orders of diffracted light interfere with each other, causing "self-interference".

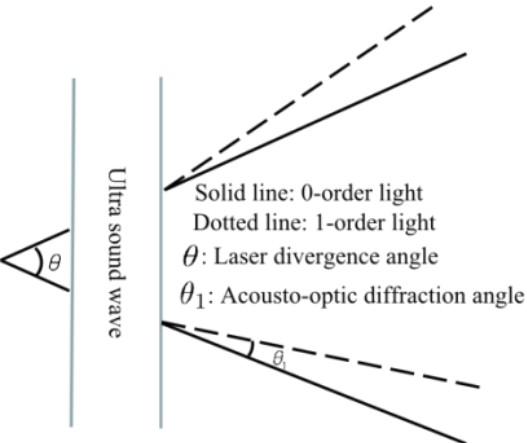

**Figure 3.** Schematic diagram of beam divergence and diffraction.

In Figure 4, we depict the detection area of the laser spot. Instead of using the traditional receiving method, we use a new amplified method to receive the optical signal. The light passing through the ultrasonic region is amplified by the convex lens, because the diffracted light angle is smaller than the beam divergence angle, so the situation shown in Figure 4 is formed, and the diffracted light is superimposed. The detector receives a part of the enlarged spot, and the 0-order and the 1-order diffracted lights are in the receiving area, and there is a frequency difference between the diffracted lights that is related to the ultrasonic frequency, thereby generating a "self-interference" signal. This receiving method can filter out unnecessary optical noise and has strong resistance to mechanical vibration.

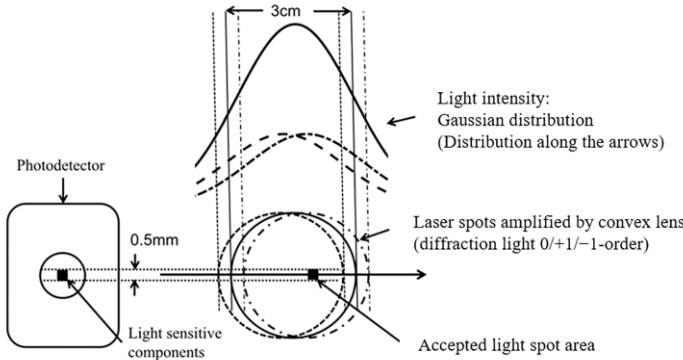

**Figure 4.** Schematic diagram of the spot detection area.

We note that the diffraction angle is inversely proportional to the wavelength of the sound wave. The sound wave's wavelength is proportionally increased as its frequency is decreased. Therefore, the diffraction angle will be reduced in proportion to the decrease in the frequency of the sound wave.

The diffracted light expression of the laser after the ultrasound is [21]:

$$
\begin{aligned}
E(t) &= \int_{-\infty}^{+\infty} E_0 \exp\left\{ jn_0 l\left[1 + m\sin\omega(t + \tfrac{y}{v})\right]\right\} \times \exp\left[j\left(\omega_0 t - \tfrac{2\pi n_0 y \sin\theta}{\lambda}\right)\right] dy \\
&= E_0 \exp(jkn_0 l) \sum_{q=-\infty}^{+\infty} J_q(kn_0 lm) \times \exp[j(\omega_0 t + q\omega)t]\delta(\tfrac{n_0 \sin\theta}{\lambda} - \tfrac{q}{\lambda_\mu})
\end{aligned}
\tag{4}
$$

where $E_0$ is the electric field of the incident light, and $k$, $\omega_0$, $\lambda$ is the wave number, angular frequency, and wavelength of the incident light. $n_0$ is the refractive index without disturbance, $m$ is the modulation constant of the refractive index, $\omega$ is the angular frequency of the ultrasonic wave, and $v$ is the sound speed. $q$ is an integer and $J_q$ contributes to the Bessel function of order $q$. $\lambda_\mu$ is the ultrasonic wavelength in the medium and $\theta$ is the diffraction angle. The symbol $l$ represents the width of the sound column, and $y$ represents the distance along the arrow, which is depicted in Figure 2.

Assuming that the diffracted light above the 2-order is weak enough to be neglected, the signal detected by the photodetector is:

$$
\begin{aligned}
I(t) &= \left| \int_{-a}^{a} dx \int_{y-d}^{y+d} dy E_0 \sum_{q=-2}^{2} J_q(kn_0 lm) \times \exp\{j|(\omega_0 + q\omega)t + kn_0 l|\} \right|^2 \\
&= 16a^2 d^2 E_0^2 \left| \sum_{q=-2}^{2} J_q(kn_0 lm) \exp\{(jq\omega t)\} \right|^2 \\
&= 16a^2 d^2 E_0^2 F(t)
\end{aligned}
\tag{5}
$$

where $F(t)$ is:

$$
\begin{aligned}
F(t) &= F(t + \tfrac{2\pi}{\omega}) \\
&= \left| \sum_{q=-2}^{2} J_q(kn_0 lm) \exp[j(q\omega t)] \right|^2 \\
&= M_0 + M_1 \sin(\omega t) + M_2 \sin(2\omega t) + M_3 \sin(3\omega t) + M_4 \sin(4\omega t)
\end{aligned}
\tag{6}
$$

As shown in Figure 5, when ultrasonic waves are transmitted to the first beam ($M_1$ to $M_2$), a "self-interference" signal is generated. The same signal as the "self-interference" signal would be generated when ultrasonic waves are transmitted to the second beam ($BS_1$ to $BS_2$). Then, we can obtain time-of-flight $P$ using the cross-correlation algorithm. When time $R_f$ is equal to the time-of-flight $P$, there will be a peak in the cross-correlation diagram. The cross-correlation formula is as follows, while $f(t)$ is the optical signal:

$$
R_f(\tau) = \int_{-\infty}^{+\infty} f(t)f(t-\tau)dt
\tag{7}
$$

The chirp signal used to drive the transducer can be represented as:

$$
F(t) = \sin(\omega_s t + \frac{\pi B}{T}t^2) \ 0 \leq t \leq T
\tag{8}
$$

$B$ is the frequency range of the chirp signal, $T$ is the pulse repetition period.

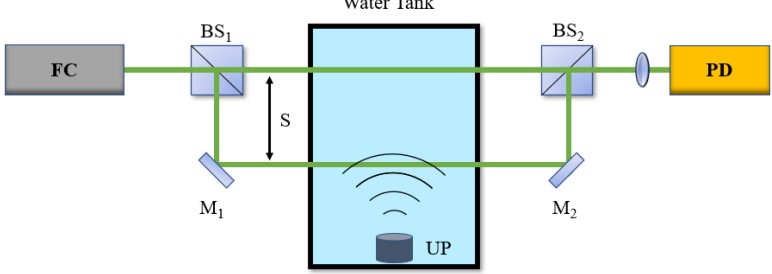

**Figure 5.** Schematic diagram of the time measurement.

*2.2. Acoustic Distance-of-Flight Measurement Principle*

Figure 6 shows the distance-of-flight measurement principle.

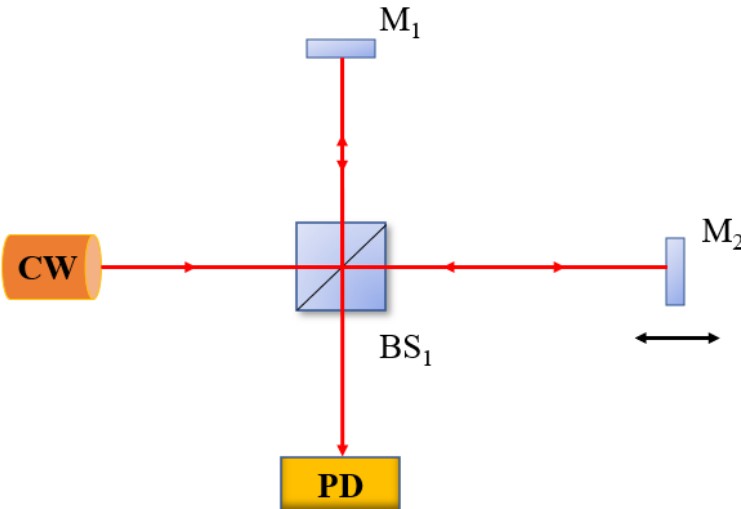

**Figure 6.** Flight distance measurement principle. CW: continuous wave laser, $M_1$-$M_2$: mirror, $BS_1$: beam splitter, PD: photodetector.

The distance-of-flight measurement is based on the continuous wave laser ranging principle. Interference occurs when two light beams have the same frequency, and the phase difference is a constant [31]. During constructive interference, the two beams are in phase and the peaks of both beams reinforce each other, resulting in a bright fringe, whereas during destructive interference the beams are out of phase and the peaks of one beam are cancelled by the troughs of the second beam, resulting in a dark fringe. By calculating the changes of the fringes, distance can be measured.

When $M_1$ is fixed while $M_2$ moves, the changing path difference results in changing interference fringes. Thus, the distance travelled by sound can be calculated using the following equation [17,32]:

$$S = (M + \Delta m)\frac{\lambda_a}{2n_g} \tag{9}$$

where $\lambda_a$ is the He-Ne laser wavelength, $n_g$ is the refractive index of air, $M$ is the integer part of the number of interference fringes, $\Delta m$ is the fractional part of the interference fringes, and $P$ is the time-of-flight. Therefore, sound speed can be calculated using the following equation:

$$V = \frac{S}{P} = (M + \Delta m)\frac{\lambda_a}{2n_g P} \tag{10}$$

*2.3. Experimental Setup*

The experimental setup for the underwater sound velocity measurement is shown in Figure 7. We used the femtosecond laser (Menlo system orange) and He-Ne laser (Thorlabs HRS015B) as our light sources. We used the linear frequency modulation signal (300 kHz–1 MHz) as the driving signal of the ultrasound probe. During the measurement of the sound speed, the frequency comb emits pulsed light (spot diameter 2 mm) and splits into two vertical pulsed light beams in $BS_1$. One is injected into the Mach–Zehnder interferometer while the other is injected into $M_3$, which is fixed on PDP (PI M-521.DD1). The output of the Mach–Zehnder interferometer combined with the scanning beam (light beam reflected on $M_3$) is injected into $PD_1$ (Thorlabs APD430A) to measure the-time-of flight, i.e., the time difference between two self-interference signals generated by optical path $M_2$ to $M_5$ and optical path $BS_3$ to $BS_4$. The filter (Mini-Circuits 15542 SLP-5+) is used to obtain the acoustic signal. An oscilloscope (Tektronix MDO3104) is used to measure and store waveforms. It is noteworthy that sound distance $S$ is one of the most crucial parameters in sound speed calculation; it is necessary to ensure that two arms of the Mach–Zehnder interferometer (optical path $M_2$–$M_5$ and optical path $BS_3$–$BS_4$) are parallel to each other. As for ultrasonic waves, we need to utilize the position-sensing detector (Thorlabs KPA101) to calibrate and modify the angle of the ultrasound probe [33], so that the transmission direction of ultrasonic waves is consistent with the optical path of $M_2$-$BS_3$ and ensures that the transmission distance of ultrasonic waves is also $S$. Due to the frequency comb characteristic, interference would occur when the light path is equal. Thus, when the scanning path ($M_3$–$M_1$–$BS_2$–$BS_5$) equals the optical path ($BS_1$–$BS_3$–$BS_4$–$BS_5$), interference would occur. Similarly, when the scanning path equals the optical path ($BS_1$–$BS_3$–$M_2$–$M_5$), interference would occur again. Therefore, the distance between two interferences, which is the optical path difference between the reference arm ($BS_1$–$BS_3$–$M_2$–$M_5$) and the measuring arm ($BS_1$–$BS_3$–$BS_4$–$BS_5$), is twice the acoustic flight distance. The distance can be obtained by the interference fringes of a continuous wave laser. During our experiment, we also used a commercial SVP (Valeport Mini SVP) to detect sound speed as a reference to verify the experimental results. We placed anechoic tiles around the water tank in the experiment to eliminate the ultrasonic echo [34]. Meanwhile, the water tank was tailored to deal with the divergence of the ultrasonic wave.

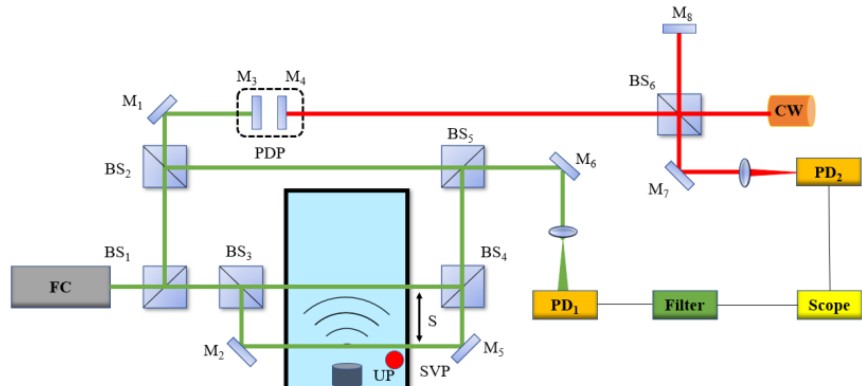

**Figure 7.** Experimental setup. FC: frequency comb, $M_1$–$M_8$: mirror, $BS_1$–$BS_6$: beam splitter, UP: ultrasound probe, $PD_1$–$PD_2$: photodetector, PDP: precision displacement platform, SVP: mini sound speed profiler.

## 3. Experimental Results and Discussion

### 3.1. Experimental Results

Figure 8a demonstrates the interference fringes of the continuous light source (He-Ne laser), and we observed that the homodyne signal varies in accordance with the sine law. The fringes of the frequency comb interferometer are depicted in Figure 8b. The cross-correlation patterns of the two bursts A and B in Figure 8a, which correspond to the two optical arms of the Mach–Zehnder interferometer, can be utilized to calculate the sound's distance-of-flight.

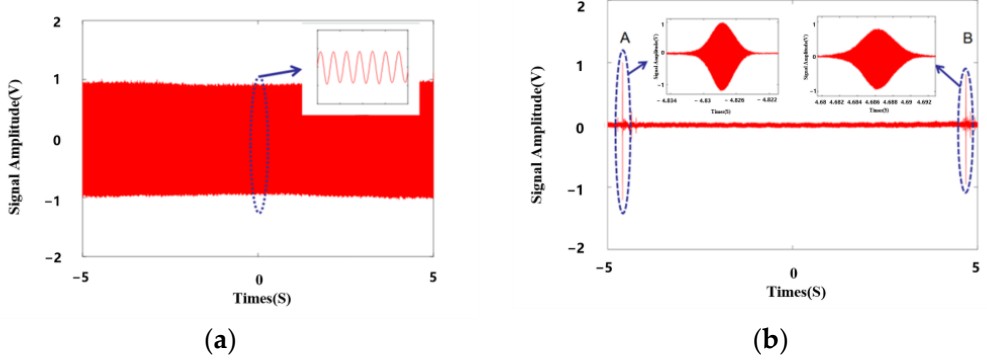

**Figure 8.** (**a**) Output signal of the He-Ne interferometer. (**b**) Femtosecond laser interference fringes caused by movement of a precision displacement stage to an equal path.

Figure 9a demonstrates the light signal resulting from the acousto-optic effect. Signals C and D represent the diffracted light interference signals generated when sound travels through the two optical paths of a Mach–Zehnder interferometer. Figure 9b depicts the result of the optical signal's cross-correlation. *P* is the required flight time. Compared with the traditional method, the utilization of a chirp highly improves waveform pulse width after cross-correlation from 40 µs to 1.5 µs.

Figure 10 shows the results over a short period; with a constant temperature system, they are highly repeatable and reproducible. The distance measurement standard deviation was 887 nm, as shown in Figure 10a. The standard deviation of the time-of-flight measurement was 1.03 ns, as shown in Figure 10b. The standard deviation of the sound speed time measurement was 0.0269 m/s, as shown in Figure 10c.

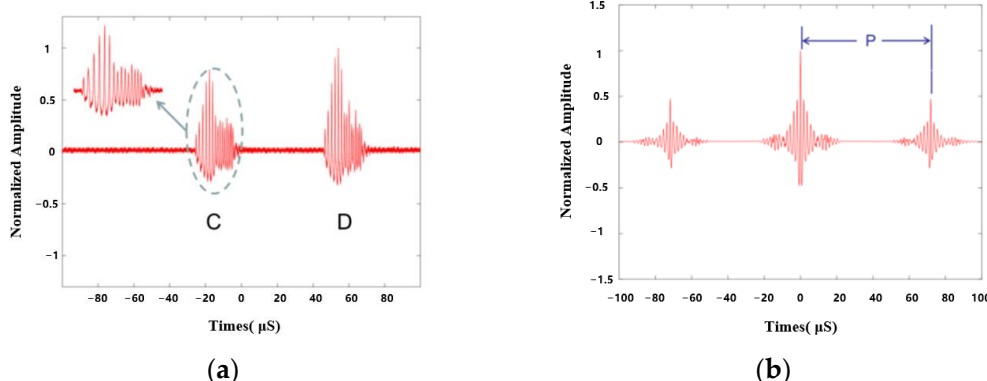

**Figure 9.** (**a**) Light signal due to the acousto-optic effect. (**b**) Signal cross-correlation diagram.

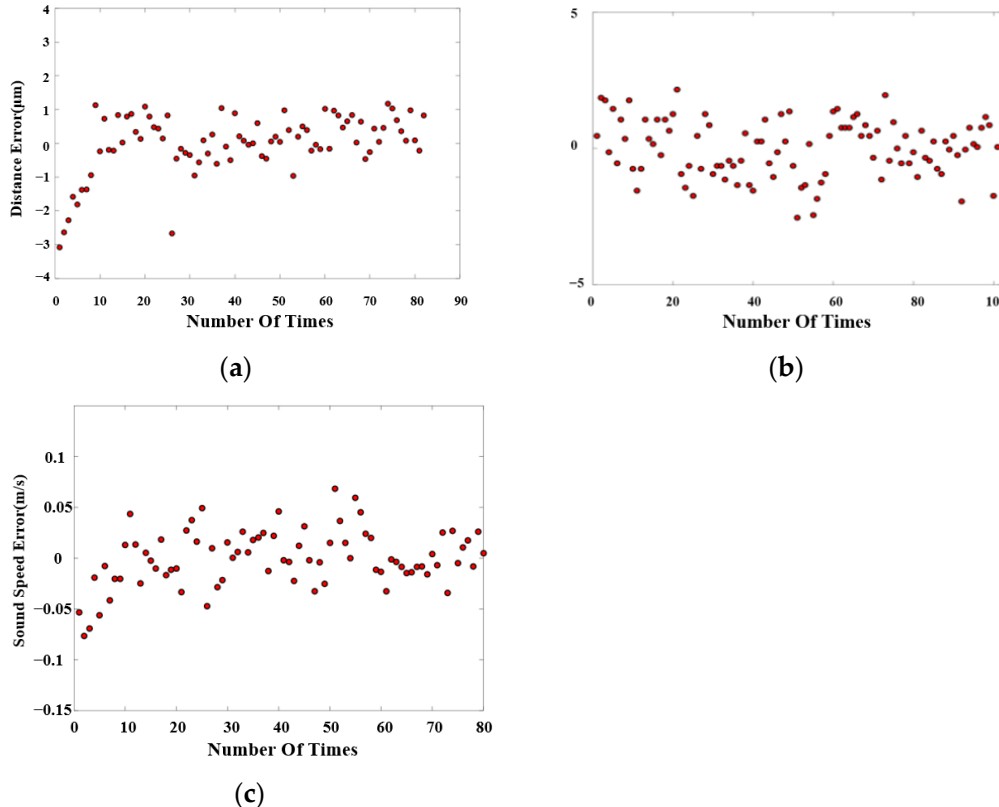

**Figure 10.** (**a**) Distance measurement results (short period). (**b**) Time-of-flight measurement results (short period). (**c**) Sound speed measurement results (short period).

During the experiments, we used the results of measurements made with a commercial sound velocity profiler (Valeport, miniSVS Sound Velocity, sampling rate of 60 Hz) as reference values for underwater sound velocity measurements. Figure 11 shows the results of this method compared to an SVP in seawater and pure water with changes in the external environment over a long period. With the variation in temperature, this method has a good follow-up.

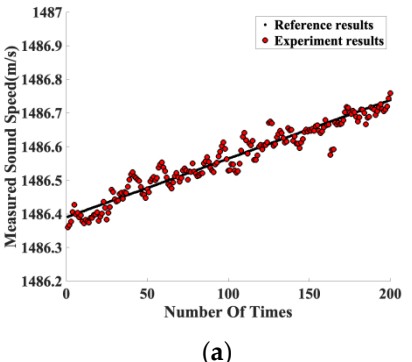 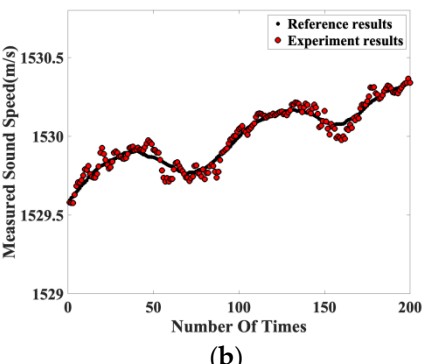

**(a)**  **(b)**

**Figure 11.** (**a**) Pure water experimental results (long term, about 1 h). (**b**) Seawater experimental results (long term, about 1 h).

### 3.2. Discussion

Compared with other methods we proposed before, a chirp signal highly improves the precision of time-of-flight and achieves a better signal-to-noise ratio. Table 1 shows the comparison of these methods, proving that the combination of "chirp signal + cross-correlation" has better performance in pulse width and time-of-flight uncertainty. With better effective bandwidth and time-of-flight uncertainty, we can apply this method in the ocean.

**Table 1.** Pulse width and TOF uncertainty of different methods.

| Method | Pulse Width | Time-of-Flight Uncertainty |
| --- | --- | --- |
| square pulse + threshold | 40 μs | 8.6 ns |
| mark signal + sing-around | / | 2.0 ns |
| chirp + cross-correlation | 1.5 μs | 1.05 ns |

This experiments also have many limitations that need to be improved. Firstly, in terms of distance measurement repeatability, we used the inter-correlation method to obtain the acoustic distance. We found that distance measurements fluctuate in the micrometer range, which is limited by the accuracy of the electric displacement table (PI M-521.DD1). The movement accuracy of the electric displacement table can only reach the micron level. This limits the accuracy of distance-of-flight measurement.

Secondly, for time-of-flight issues, the oscilloscope (Tektronix MDO3104) we used will introduce an error of 2 ns or less. Furthermore, we experimented with a signal sampling rate of 1 Ghz (the subsequent algorithm interpolates the original signal), as shown in Figure 12. We can see that time-of-flight is discrete and limited by the sampling rate. In addition, we used a femtosecond optical frequency comb with a repetition frequency of 100 MHz, which is expressed in the time domain as one optical pulse every 10 ns. We can interpret this as optically sampling the sound field every 10 ns. (The signal is subsequently passed through a filter (Mini-Circuits 15542 SLP-5+) to obtain a signal with the same frequency as the sound field.) Both the sampling rate and the accuracy of the oscilloscope affect the repeatability of time-of-flight. Therefore, the resolution of the speed of sound will be affected.

Thirdly, due to site constraints, it is difficult for us to ensure that the beams of Mach–Zehnder interference are highly parallel. Non-parallelism of the beam can lead to some deviation of the measurement results from the actual sound distance. Ignoring this, the time is well reproducible. From what we can see, the potential of this method is a benchmark for sound speed measurement.

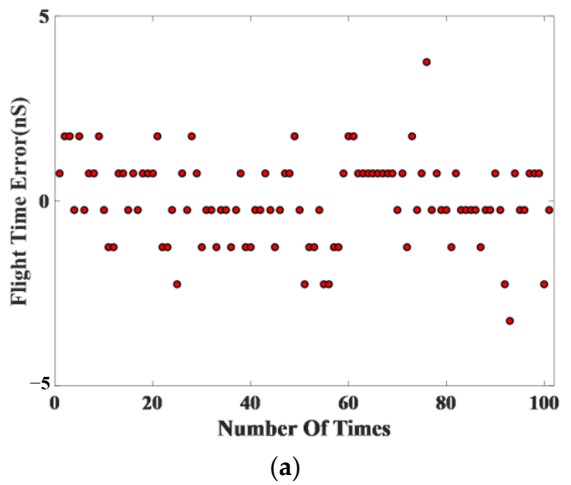
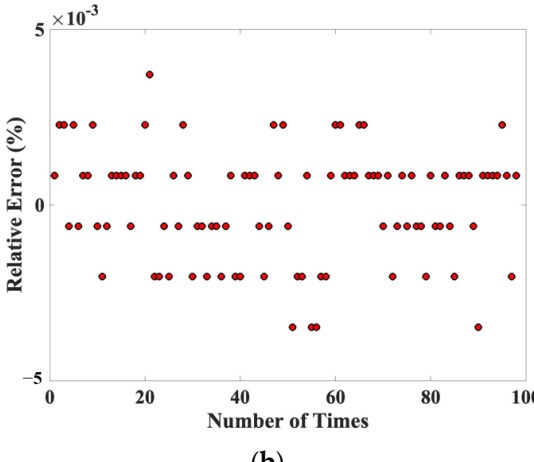

(**a**)  (**b**)

**Figure 12.** (**a**) Time-of-flight measurement results (without interpolation). (**b**) Relative error of time-of-flight.

All the above problems can be solved by upgrading the hardware. We can reduce costs and obtain a more accurate distance measurement by replacing the femtosecond optical frequency comb with a dual-frequency laser, which is a proven technology. For time-of-flight, we can connect the acquisition card clock to a rubidium clock (namely, so that the whole system uses a common time base) and use an acquisition card with a higher sampling rate.

### 4. Uncertainty Analysis of Sound Speed

There is uncertainty in the Ciddor formula itself and in the parameters within the formula. As shown in Table 1, uncertainty in the speed of sound from these causes is too small ($10^{-7} \cdot v$ m/s) to be negligible compared to the above. The specific respective uncertainties are also given in Table 1.

From Equation (1), the uncertainty of the measurement result can be calculated as:

$$
\begin{aligned}
u_v &= \sqrt{\left(\frac{\delta v}{\delta S} \cdot u_s\right)^2 + \left(\frac{\delta v}{\delta P} \cdot u_p\right)^2} \\
&= \sqrt{\left(\frac{1}{P} \cdot u_s\right)^2 + \left(\frac{S}{P^2} \cdot u_p\right)^2}
\end{aligned}
\tag{11}
$$

The first term of Equation (11) represents uncertainty in the speed of sound due to the distance measurement, and the second term represents uncertainty in the speed of sound due to the time-of-flight measurement. We have analyzed the uncertainties for distance and time-of-flight above, so we can obtain the synthetic uncertainty for the speed of sound. As in Table 2, the synthetic uncertainty is 0.0269 m/s.

**Table 2.** Contribution of the above factors to the uncertainty.

| Sources of Measurement Uncertainty | Uncertainty | Uncertainty Value |
|---|---|---|
| Distance | 887 nm | 0.0128 m/s |
| Due to air refractive index | $10^{-7}$ v m/s | $1.47 \times 10^{-4}$ m/s |
| Environmental temperature uncertainty | 25 mK | |
| Environmental air pressure uncertainty | 15 pa | |
| Environmental humidity uncertainty | 2% | |
| Due to Ciddor formula | $2 \times 10^{-8}$ | |
| Time-of-flight | 1.03 ns | 0.0225 m/s |
| Combined uncertainty | $[(0.0128)^2 + (0.0225)^2 + (1.47 \times 10^{-4})^2]^{1/2}$ | 0.0269 m/s |

## 5. Conclusions

In this study, we completed a sound speed measurement based on the acousto-optic self-interference effect between sound and an optical frequency comb. The acousto-optical self-interference effect between sound and an optical frequency comb allows rubidium microwave frequency scales to be linked to sound speed standards in a precise and simple way, providing a vehicle for the development of high-resolution and high-accuracy sound speed standards. This allows us to trace the sound speed error back to Rubidium clock standards. In addition, the method proposed in this study is more in line with the definition of sound speed since distance and time can both be obtained by an optical frequency comb. The results shown in this study have good repeatability. Uncertainty can reach the level of 0.0269 m/s. Our experiment is limited by the instruments used, which can be seen in the discrete results of time-of-flight. The displacement table we used has an accuracy of 1 μm, which also affects the accuracy of our distance measurements. In conclusion, this method can reach a higher accuracy of sound speed and has the potential to be the standard of sound speed.

Future experiments should first consider the feasibility of the acousto-optic method. Currently, the measurement setup is expensive and large in scale, which makes it difficult to put it in the ocean for a long time. Using an acquisition card and dual-frequency laser might be an option to simplify the setup. Automation of the setup might improve feasibility; future research might apply an automatic measuring setup based on this method. Then, we can improve the accuracy of measurements by altering the optical frequency comb to a dual-comb laser to achieve better uncertainty in distance-of-flight measurement. More sensors can be applied to obtain information such as temperature, salt, and depth to achieve real-time calibration.

**Author Contributions:** Conceptualization, Z.Y. and F.D.; data curation, H.L. and Z.L.; funding acquisition, B.X.; methodology, Z.Y. and F.D.; project administration, H.L. and B.X.; supervision, H.L. and B.X.; writing—original draft, Z.Y. and F.D.; writing—review and editing, X.Y. All authors have read and agreed to the published version of the manuscript.

**Funding:** This work is supported by National Natural Science Foundation of China (No. 62075162, No. 62001329), Natural Science Foundation of Tianjin City (No. 19JCQNJC01700), State Key Laboratory of precision measuring technology and instruments (pilab2104).

**Institutional Review Board Statement:** This study did not require ethical approval.

**Informed Consent Statement:** Not applicable.

**Data Availability Statement:** Not applicable.

**Conflicts of Interest:** The authors declare no conflict of interest.

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
