# Peer review of "Direct Underwater Sound Velocity Measurement Based on the Acousto-Optic Self-Interference Effect between the Chirp Signal and the Optical Frequency Comb"

_jmse, doi:10.3390/jmse11010018_

Round 1

Reviewer 1 Report

1. All the references are not cited e.g. 25

2. What is the feasibility of the setup for an underwater vehicle? The proposed setup was tested on a water tank only.

3. referencing and citation are not proper. some more references may be added for better support of the proposed method.

Reviewer 2 Report

Review of

Direct underwater sound velocity measurement-based
acousto-optic self-interference effect between a chirp signal3
and optical frequency comb

Authors

Zihui Yang, Fanpeng Dong, Hongguang Liu, Xiaoxia Yang, Bin Xue

Abstract

The Authors propose a new method to measure under-water velocity using acousto optics and claim it has high precision.

REVIEWER COMMENTS

The paper is not clear and also not very well written since in many parts the English grammar is wrong, and in many cases the symbols used in the equations are not introduced in the text, or the figures are not well described. Sometimes, the wording used is not scientifically correct.

I suggest the authors re-write the paper in a clear way before submitting it again.

The paper seems to be very interesting but needs to be written using better grammar and needs to be more clear.

In the following I present my comments on the different sections of the paper:

1 Introduction

The Authors say that they discovered the "acousto-optic self-interference effect" that is an effect that happens when an ultrasonic wave interacts with an optical frequency comb. It is not clear what self-interference means in this case! Is it the diffracted light that interferes with itself?

Then the authors continue saying: "In this study, we first discovered that diffracted light at each level interferes with each other when light travels through an ultrasonically perturbed material and causes diffraction."

First of all the grammar of this sentence is not correct in english, and second of all there are a series of mistakes which make the underlying physics unclear.

  • it is not clear what is meant by diffracted light at each level interferes with each other! Maybe they mean that different orders of diffracted light interfere with each other? Or they mean interference of each order with itself since before they were talking about self interference?

  • The second part of the sentence is also wrong: light does not cause diffraction! Maybe the Authors mean that light undergoes diffraction?

Continuing on the same paragraph, the Authors mention "the acousto-optic self-interference signal" and say it has similar characteristics as "the acousto-optic signal". First of all what is an acousto-optic signal? In acousto-optics there is an acoustic signal and an optical signal but not an acousto-optical signal! There is an acousto-optical interaction where an acoustic signal or wave causes diffraction of an optical signal or wave generating a diffracted signal or wave. Could it be that the authors call the diffracted light, the acousto-optical signal? The Authors should also describe what they mean by an acousto-optic self-interference signal. Maybe they mean the diffracted wave (or signal) that interferes with itself?

The nomenclature used by the Authors is not very cear, and my suggestion is that besides mproving the english grammar of this paper the authors also use a better nomenclature.

2 Measuring principle and experimental setup

The Authors say that the path difference in the Mach-Zehnder interferometer is twice the sound distance S. But the sound distance S is not shown in the fugure! Furthermore sound does not have a distance! Maybe the Authors mean the distance travelled by the sound wave? Is this the distance between the two arms of the interferometer? If so then it would be a good idea to show it in the figure and say what it is in the text.

    1. Principle for time-of-flight measurement

      Also here the Authors talk about "levels" of light but maybe they mean orders?

      They say that: "diffracted light at each level interferes with each other, at whici point the interference signal is an integer multiple of the acoustic frequency and can be detected by the photodetector".

      The grammar of the first part of the sentence is wrong as was the case in the introduction. Whar does interferes with each other mean?

      Furthermore, the claim that an interference signal is a multiple of a frequency is not very clear or very well described from a physical point of view. Is it the frequency of this signal being a multple? Of is it the amplitude being a multiple? The authors need to describe what is happening in a clear way!

      The rest of the paragraph is also not clear! The Authors say:"when the sound travels between the the reference arm and the measurement arm the detector receives the same signal". Which detector? The setup was never described! What same signal? Same of what?

      The way the authors describe their measurement is not very clear. I feel that it is only a flow of seemingly complicated words without a clear structure for the meassage the authrs want to deliver!

      The description of figure 2 is acceptable, but the description of figure 3 is not! The Authors say that figure 3 depicts the detection area of the laser spot, but on this figure there are many things: circles, peaks and distances that are not described!

      In Eq(2) for the diffraction angle the authors do not say what n is. Furthermore, what the authors call "the ultrasonic wavelength of the medium" has two different symbols in the text and in the Eq(2). Second of all, a medium does not have a wavelength. Maybe the Authors mean the ultrasonic wavelength in the medium (and not of the medium)?

      There is no descrption of Eq(4) and of Eq(5).

      The the authors say "as stated above, when sound passes...". Here, in reality there was nothing stated above! The setup shown in Fig. 1 was never described and the case of parallel paths was never described!

      In the text the Authors make use of symbol P to indicate time-of-flight but shortly after they call it Rf in Eq(6). In Eq(7) the authors do not say what B is.

    2. Acoustic distance-of-flight measurement principle

      In Fig.4 the authors say theu show the distance-of-flight measurement principle but in reality they only show a Michelson interferemoter. There is no sound wave depicted in this figure. In the description f the figure the authors never mention the sound wave. They only describe a simple interference phenomenon but never say how the sound wave comes into this measurement as if the reader already knows! At the end of the paragraph the authors for the first time give an equation to measure something called "sound range" without saying what it is! Then they give Eq(9) to calculate the sound speed but also here without describing one of the symbols used. In particular symbol P is not described.

    3. Experimental setup

      Finally the Authors present in Fig.5 the experimental setup. But the description of the setup is not clear! They say that: "during detection, the frequency comb emits pulsed light and splits into two vertical lights in BS1.

      - what is detection? Id detection a phase of the measurment? Are there other phases? The authors should describe things in a clear way.

      -what are vertical lights? Do the authors mean vertically polarized light beams?

      The authors say the M3 is fixed on PHP. But what is PHP? In Fig.5 there is a symbol PDP. Are these the same thing? What is this?

      The Authors say:"due to the ultrasonic wave propagation of the optical path". This sentence has no meaning in english! Maybe the authors mean ultrasonic wave propagation accross the optical path?

The authors describe the setup but do not say the properties f the optical beam. How large is the beam in the tank?

The authors talk about sound dstance but never show it in the figure! Is this the distance travelled by sound through the tank? Or is it the distance between the two light beams, that is also travelled by the sound wave? Furthermore, the term sound distance is not scientifically correct!

Reviewer 3 Report

First of all, I want to congratulate the authors for their efforts in this manuscript. The paper analyses the direct underwater sound velocity measurement-based acousto-optic self-interference effect between a chirp signal and optical frequency comb. The topic is aligned with the journal scope and is interesting for the readers. There are some aspects to be improved before accepting the paper. One of the major limitations of the work is the lack of an extensive review of the existing similar experiments and the lack of context, with references, for readers not experts on the importance of sound. Following, I include the list of issues to be corrected:

Major changes:

1.       In the introduction, the authors have to contextualise the use of underwater sound and their main uses, such as for communication (10.1109/JSEN.2015.2434890), object detection (10.1109/OCEANSChennai45887.2022.9775385), etc.

2.       Authors must add a related work section in order to highlight the gap in the current technologies and demonstrate the novelty of their study. In the related work, a summary of published papers analysing the same of the similar topic should be presented.

3.       In the results, the error is given as an absolute error. It is recommended to include data about the relative error too.

4.       After resenting the results, a discussion about comparing the results with papers included in related work must be added. Limitations of the experiments and the impact of the research in marine science should be highlighted.

5.       At the end of conclusions, the future work has to be added in a new paragraph.

Minor changes:

Consider avoiding using the same words in the abstract and in the title.

Change combs.[11] à combs [11]. Check other similar cases

Round 2

Reviewer 3 Report

All my comments are correctly addressed, and the paper has been improved.